# Polyesters as a Model System for Building Primitive Biologies from Non-Biological Prebiotic Chemistry

**DOI:** 10.3390/life10010006

**Published:** 2020-01-19

**Authors:** Kuhan Chandru, Irena Mamajanov, H. James Cleaves, Tony Z. Jia

**Affiliations:** 1Space Science Center (ANGKASA), Institute of Climate Change, Level 3, Research Complex, National University of Malaysia, UKM Bangi, Selangor 43600, Malaysia; 2Department of Physical Chemistry, University of Chemistry and Technology, Prague, Technicka 5, Dejvice, 16628 Prague 6, Czech Republic; 3Earth-Life Science Institute, Tokyo Institute of Technology, 2-12-1-IE-1 Ookayama, Meguro-ku, Tokyo 152-8550, Japan; irena.mamajanov@elsi.jp (I.M.); henderson.cleaves@gmail.com (H.J.C.II); 4Blue Marble Space Institute for Science, 1001 4th Ave, Suite 3201, Seattle, WA 98154, USA; 5Institute for Advanced Study, Princeton, NJ 08540, USA; 6Center for Chemical Evolution, Georgia Institute of Technology, Atlanta, GA 30332, USA

**Keywords:** polyesters, origins of life, non-biomolecules, prebiotic chemistry, wet-dry cycles, protocells

## Abstract

A variety of organic chemicals were likely available on prebiotic Earth. These derived from diverse processes including atmospheric and geochemical synthesis and extraterrestrial input, and were delivered to environments including oceans, lakes, and subaerial hot springs. Prebiotic chemistry generates both molecules used by modern organisms, such as proteinaceous amino acids, as well as many molecule types not used in biochemistry. As prebiotic chemical diversity was likely high, and the core of biochemistry uses a rather small set of common building blocks, the majority of prebiotically available organic compounds may not have been those used in modern biochemistry. Chemical evolution was unlikely to have been able to discriminate which molecules would eventually be used in biology, and instead, interactions among compounds were governed simply by abundance and chemical reactivity. Previous work has shown that likely prebiotically available α-hydroxy acids can combinatorially polymerize into polyesters that self-assemble to create new phases which are able to compartmentalize other molecule types. The unexpectedly rich complexity of hydroxy acid chemistry and the likely enormous structural diversity of prebiotic organic chemistry suggests chemical evolution could have been heavily influenced by molecules not used in contemporary biochemistry, and that there is a considerable amount of prebiotic chemistry which remains unexplored.

## 1. Introduction 

How life emerged from inanimate chemistry is an unresolved scientific question and an area of active research [1,2,3]. Contemporary biochemistry uses a relatively small set of building blocks, e.g., only 20 coded amino acids, four ribonucleotides, four deoxyribonucleotides, etc., out of a large number of possibilities [4]. We here term the molecules used in contemporary biochemistry “biomolecules” to distinguish them from “non-biomolecules,” which are molecules not used for major macromolecular functions in contemporary biochemistry (or even not used at all in contemporary biochemistry), but could have played much more important roles in previous biological or proto-biological states. How, why, and at what stage of chemical and/or biological evolution the selection and canonicalization of the biomolecules occurred remains unclear. Much of origins of life (OoL) research is oriented towards synthesizing or reproducing biomolecules or processes under presumed primitive Earth conditions [2]. While such approaches are logical given the evidence available from contemporary biology, and appear to be a sound application of “Occam’s razor,” they are also often based on a number of assumptions. One such assumption is the direct continuity between biochemistry and prebiotic chemistry, which has given rise to OoL models variously focused on the primary roles of lipids [5], nucleic acids [6], metabolism [7], or proteins [8]. However, contemporary biomolecule-based hypotheses overlook the long-recognized challenges of robust prebiotic syntheses of biomolecules [9], and the fact that many abiotic syntheses produce a tremendous diversity of organic compounds. 

For example, researchers often consider carbonaceous chondrite meteorites to be proxies that validate laboratory prebiotic chemistry simulations [10,11]. Carbonaceous meteorites contain a myriad of organic compounds divided between a soluble fraction and an insoluble organic matter (IOM) phase consisting of intractable complex macromolecular organic material. The soluble fraction is often incredibly diverse, for example 14,000–50,000 unique molecular formula compounds were identified in a methanolic extract of the Murchison meteorite, which it was estimated may represent millions of unique structural isomers [12] (Figure 1). 

In fact, the number and diversity of compounds in the Murchison meteorite can only be crudely estimated due to analytical challenges [13], suggesting that the actual organic complexity may be far greater. The classes of molecules identified to date in the Murchison meteorite include amino acids [14], hydroxy acids [15], carboxylic acids [16], and nitrogen heterocycles [17]. Most of these classes are dominated by compounds not used in present-day biology [18]. In light of this, it seems plausible that a similarly complex compound suite was provided to primitive Earth (or existed on another planet [19]) from one or more sources (e.g., [20,21,22]), and indeed various prebiotic chemistry simulations similarly produce a complex array of organic products (Figure 2) [23].

Given the magnitude of prebiotic chemical diversity, the early environment likely included non-biomolecules along with biomolecules and quite possibly more of the former than the latter. However, relatively little effort has been expended towards understanding the identities or reactivity of the non-biomolecules generated in prebiotic syntheses [24], or the processes or novel chemical phenomena they might enable. It may be likely that during early chemical evolution, proto-biological functions involved different-possibly larger, smaller, or partially overlapping sets of chemical compounds compared to contemporary biochemistry.

Although several studies suggest that biomolecules conferred a significant advantage to primitive systems resulting in their dominance in contemporary biochemistry [25,26,27] it does not necessarily follow that life was originally based on the same biomolecules. Indeed, it is known that biomolecules represent a minute fraction of the molecular diversity produced by prebiotic chemistry (as mentioned above) and this would tend to lower the probability that chemical bonds formed exclusively among biomolecules particularly in a one-pot prebiotic synthesis [3,24,28]. The possible importance of non-biomolecules in the OoL has been suggested previously as a sort of “chemical opportunism” [29]. According to this notion whatever molecules conferred the greatest selective advantage (via function, persistence, availability, etc.) would be selected by primitive chemical systems. If non-biomolecules conferred a greater advantage to primitive chemical systems over biomolecules, then non-biomolecules would have dominated. A notional example of such “chemical opportunism” is the emergent systemic autocatalysis proposed by [30]. The idea that the first autocatalytic chemical systems could have been composed of non-biomolecules has also been proposed by Lancet and coworkers [19,29]. More nuanced examples of primitive non-biomolecular systems, include those incorporating purely in vitro-produced polymers (themselves produced from biomolecular components such as amino acids) which do not participate in modern biology, have also been explored including self-assemblies of lipid-like peptides [31]. Earlier examples of non-biomolecule-based OoL models include the so-called pre-RNA world [32,33,34,35] and polyester-based origins of life models [36]. 

Polyesters offer a particularly interesting model for studying chemical evolution due to their structural similarity to peptides, ease of polymerization into diverse polydisperse polyester mixtures (polyesters form easily during low-temperature wet-dry cycling of hydroxy acids [37,38]) (Figure 3), and ability to perform enzyme-like catalysis [39] and compartmentalization of molecules through microdroplet assembly [40]. Polyester-based oligomers are fundamentally different from contemporary biological macromolecular catalysts such as proteins. Polyester systems can interact due to hydrophilic, hydrophobic, and electrostatic interactions such as dipole-dipole interactions [41], but they lack the ability to form regular repeating intramolecular hydrogen bonding motifs of the sort that facilitate protein folding [42,43]. However, although polyesters cannot structurally fold in the same way as polypeptides, they might still be distant relatives of biochemistry and could have provided functions that promoted primitive molecular assembly and dynamics essential for the emergence of biology. 

Simple non-biomolecule monomers, best experimentally demonstrated so far using α-hydroxy acids (αHAs) [38,39,40,44], but which could include many other prebiotic monomer types, can form diverse macromolecular assemblages with a wide range of physical properties and functions such as self-assembly and catalysis. Though these are often dismissively lumped together as “tars,” such assemblies could present opportunities to form extremely complex “ecosystems” of microcompartments, each with different chemical properties, in which different abiotic chemistries could have been facilitated and their products exchanged. These could represent a pre-genomic analog of Woese’s rampant horizontal genome transfer-characterized LUCA [45], also known as a progenote, and an important aspect of global chemical evolution. According to this model, prior to LUCA, a community of progenotes [45], as in primitive entities in the process of (but prior to) establishing the genotype-phenotype relationship, could have dominated the prebiotic world while allowing for genetic information transfer amongst themselves and the surrounding environment. These informational materials may not be limited to nucleic acid-based materials, but also could be composed of other types of genetic information, such as those based on the composome model [29]. Such a character could have arisen among such systems, and then spread throughout them, co-opting and replacing their more primitive modes of reproduction, such as autocatalytic chemical sets proposed by Kaufman [30], potentially eventually leading to systems with established genotype-phenotype relationships that could have evolved into modern biology as-we-know-it.

## 2. Polyesters as a Model System for Chemical Evolution

The free energy of the peptide bond is around +3.5 kcal mol^−1^ under physiological conditions [46]. In biochemistry, peptides are synthesized from amino acids through a complex system of enzyme-mediated, energy-consuming reactions. Abiotic peptide synthesis is often accomplished with varying degrees of success through activation (e.g., [47,48,49,50]), heating, and/or low water activity conditions (e.g., [51,52,53]). Alternatively, polyesters, which have a considerably lower energy for bond formation (~0 kcal mol^−1^ [54]), have been considered as possible ancestral precursors of peptides [36,55]. Such polyesters can be easily produced by drying hydroxy acids, the monomer precursors of polyesters. The loss of water, for example during evaporative drying, shifts the equilibrium towards polyester elongation [38] (Scheme 1). Hydroxy acids are equally as prebiotically plausible as amino acids, as hydroxy acids may be produced in the same model prebiotic reactions that produce amino acids [56,57], and are also common in meteorites [58].

Esters are common in modern biochemistry. For example, many cell membrane lipids are esters of glycerol, phosphate, and fatty acids. Another example is cutin, a polyester synthesized from ω-hydroxy acids that is the main component of the cuticle that covers plant aerial surfaces [59]. Polyhydroxyalkanoates are also produced by various bacteria as energy-storage molecules [60], and pores assembled from polyphosphate associations with β-hydroxy acid polymers have been isolated from *E. coli* [61].

While the relative lability of polyesters could be considered a disadvantage, the reversibility of ester formation provides an experimental model to study dynamic synthesis/hydrolysis cycling, potentially caused by environmental perturbations such as seasonal or diurnal cycling. This property makes polyester systems an abstract model of non-biomolecular prebiotic polymer formation suitable for the study of the environmental selection of different chemical properties. For example, the abiotic reactions between αHA and α-amino acid monomers have been studied in model wet-dry cycling experiments [44]. These co-polymerization reactions initially produce depsipeptide oligomers that contain both ester and amide linkages. Over time oligomers enriched in peptide bonds are generated through a combination of ester–amide bond exchange and ester bond hydrolysis reactions [44]. This is a good example of how a plausibly prebiotic non-biomolecule could help scaffold the formation of a biomolecule type in a primitive environment.

We have previously characterized the polymerization of L-malic acid, a dicarboxylic αHA, by wet-dry cycling simulating various geological settings that dry intermittently [37]. In this system, L-malic acid polymerizes during drying and hydrolyzes during rehydration. If the wet phase temperature in these reactions is sufficiently low, the polymer formed in the dry phase is kinetically trapped and undergoes only partial hydrolysis, suggesting that some components of the product mixture, i.e., the most stable/persistent products, could be carried over from cycle to cycle. The presence of carryover products thus raises the possibility of selective preservation and amplification of specific polymeric structures or functionalities, i.e., stability towards hydrolysis due to intramolecular folding or intermolecular aggregation. 

Primitive polymerization processes should ideally survey large swathes of sequence space to effectively find the most stable or catalytic sequences. If a wide range of sequence space is not accessible, it is possible that the most stable and/or catalytic polymers cannot be produced. Reversible esterification offers a facile way to access polymer sequence space. Taking advantage of favorable polymerization thermodynamics, starting from a pool of 5 different αHA monomer types, a large diversity of oligomer sequences was achieved (Figure 3) [38]. The experiments in this study generated vast, likely sequence-complete libraries over a variety of reaction conditions (temperature, concentration, salinity, and presence of congeners) compatible with geochemical settings on primitive Earth and in other Solar System environments.

While many biopolymers, i.e., proteins and nucleic acids, are linear polymers (as was the model polyester system studied in [38], for purely analytical tractability purposes), some biopolymers, i.e., glycogen, amylopectin, and dextran, are branched. In abiotic settings, diverse pools of monomers likely oligomerized without enzymatic control, forming significant amounts of branched and crosslinked polymers, further increasing the chemical complexity above what could be present with purely linear polymers. Such branched polymers have been considered as possible precursors to globular proteins [36] and enzymes [39]. When branched polyesters are generated under wet/dry cycling or continuous drying conditions, different properties may be “imprinted” on the products [62]. For example, periodic wetting, during which partial hydrolysis of the polymer may occur, inhibits polymerization and helping keep the oligomer products water soluble which is presumably important for their function. Continuous drying, on the other hand, favors the production of insoluble high molecular weight crosslinked polyester structures [62].

While we have argued that the prebiotic milieu (for example, chemicals that are produced in model prebiotic reactions such as spark discharge experiments or those found in carbonaceous meteorites) is a “messy” chemical system with high molecular diversity, identifying each and every compound in such a system is still beyond the abilities of contemporary analytical chemistry [13]. The combinatorial hydroxy acid systems that have discussed here could be combinatorially complex, but likely still compositionally simple compared to what might be expected in a real environmental setting. Further advancement of analytical methods and techniques will allow future studies to examine systems including still greater chemical diversity, including both non-biomolecules and biomolecules.

## 3. Self-Assembled Polyester Microdroplet Compartments

Segregation and compartmentalization are fundamental cellular functions, and these can be achieved even with simple membraneless polyester droplets [40]. In the case of polyesters, drying aqueous solutions of mixed αHAs at a mild temperature (80 °C) produces a combinatorially diverse set of oligomers, which upon rehydration form gels that self-assemble into membraneless droplets (Figure 4). These droplets differentially segregate and compartmentalize various hydrophobic organic dyes and fluorescently labeled nucleic acids [40], suggesting a potential function as primitive compartments. Such droplets, or other proposed prebiotic compartments with similar functionality such as liquid-liquid phase-separated membraneless coacervate droplets [63,64,65] or lipid vesicles [5], may have served as protocells.

The polyester microdroplets generated in our studies likely assemble through hydrophobic interactions, and their surfaces and interiors are also likely non-polar, while their interiors contain less water than the surrounding aqueous environment [40]. Such microdroplets may have provided non-polar environments important for the function, synthesis, or accumulation of certain molecules. In a system with coexisting phases with differing chemical character, a given molecule could have a different affinity for each phase. The ratio of the amount of molecule that dissolves in one phase versus the other is known as the partition coefficient [66]. The partition coefficient largely depends on physicochemical properties of both the solute in question and the two phases it partitions into, and can be modulated by changing the charge, polarity, crowding, etc. of the components. In the case of non-polar polyester droplets in water, the partition coefficient determines how much of a molecule becomes sequestered in the polyester droplets. Non-polar molecules such as a hydrophobic fluorescent dye like thioflavin T tend to segregate within non-polar polyester droplets [40]. The amount of accumulation of thioflavin T within the droplets is correlated to the partition coefficient and inversely correlated to the relative volume of the droplet phase versus that of the aqueous phase, resulting in a specific concentration increase of the solute in the droplet phase. As another example, the structures of proteins change depending on solvent polarity, and would be expected to change within polyester droplets compared to polar solvents [67,68]. Likewise, primitive peptides containing large amounts of hydrophobic residues would likely have different structures (and by extension, functions) in an aqueous environment compared to a non-aqueous environment such as the droplet interiors. The interfaces between different polarity phases, such as those between polyester microdroplets or supercritical carbon dioxide droplets [69] and water, have also been shown to facilitate lipid assembly [40,70] and accumulation of amphiphilic peptides [70]. Such assembly or accumulation mechanisms could be extended to other possibly important early self-assembly chemistries [71].

The ability of polyester microdroplets to segregate and compartmentalize various molecules and reactions with different chemical characteristics could be useful for enabling higher-order chemical phenomena (Figure 5). As a hypothetical example, during template-directed non-enzymatic RNA polymerization in solution using activated nucleotide monomers [72,73], hydrolyzed, deactivated monomers accumulate over time, and these hinder template-directed polymerization by competitively occupying the complementary sites on the template strand [72]. Removal or reactivation of these oligomerization inhibitors is an important unresolved problem [74]. Potentially, conducting template-directed polymerization in the presence of droplet phases with an affinity for deactivated monomers could provide a sink for these inhibitory waste products.

Although polyester droplets could be engineered to display interesting chemical properties, whether the droplets themselves were direct or earlier transitional precursors to modern cells (and how this transition occurred) is an open question. Polyester droplets can scaffold the assembly of lipids around themselves, resulting in a molecularly crowded condensed phase with a lipid boundary [40]. Assembly of such a lipid encapsulated condensed-phases has been demonstrated with various membraneless droplet systems such as coacervates [63,64]. Encapsulating layers could also be composed of fatty acids, phospholipids, or even other molecules such as detergents [75], amphiphilic peptides [76], DNA liquid crystals [77], or even mineral particles [78]. 

Finally, polyester microdroplets can segregate fluorescently labeled nucleic acids and fluorescent proteins [40]. Primitive cells can, perhaps over-simplistically, be conceived of as molecularly crowded droplets containing genetic or functional polymers encased by a lipid boundary [79]. This is a reductionist view that obviously ignores the role lipids play in generating the cytosolic contents, and the role the cytosolic contents play in generating the cell membrane in modern biology. Nevertheless, that a cell-like lipid-enclosed droplet containing functional polymers [40] can be assembled using non-biomolecules suggests that much remains to be learned about the functional structures that may have been important for the emergence of life.

The study mentioned above [40] only included a small number of αHAs monomers, chosen for analytical tractability, which could only form linear polymers and was simply a proof-of-principle investigation that did not explore the effect of the presence of much greater molecular diversity [15]). However, by increasing the chemical diversity of αHAs such as by varying compositional ratios and introducing other non-amino-acid analog αHAs, the ability for a polymeric system to assemble into such droplets should still remain as long as the synthesized polyesters are of sufficient length (so they are insoluble in water) and that the initial monomer composition is not composed of a large quantity of very soluble species (such as polar monomers, charged monomers, or glycolic acid). In fact, by increasing the diversity of ester-bond forming monomers [18,80], a wider variety of higher-order chemistries could be obtained from droplets of this type. In addition to polyester droplets with non-polar interiors [40], polyester droplets with other interior properties, e.g., polar or charged interiors, could have also assembled depending on their chemical composition. Droplets could also co-assemble with other non-biomolecular chemistries or assemblies. Such complex polyester droplets could have acted as microreactors able to concentrate primitive chemical reactants. While we did not perform any functional searches for catalytic behavior within the droplets, incorporating hyperbranched structures [39], additional functional groups (e.g., charged groups), or other prebiotically available biomolecules (e.g., amino acids) into the droplet composition could result in potential catalytic systems and are a target of current investigations.

## 4. Prospective

After characterization of a more complex polyester system, the next step would be to increase chemical diversity to include other chemicals that would have likely existed in the same prebiotic milieu. Such realistic chemical diversity would likely result in systems which are difficult to characterize, except with respect to systemic functional or bulk properties. There are a few laboratory simulations of such stepwise increases in chemical diversity (examples include recent work with depsipeptides [44] and DNA/RNA chimeras [81]) but these studies are just the first step towards laboratory simulations of truly complex chemical systems that rival the actual complexity of the prebiotic milieu. It may actually be the case that such extremely chemically complex and heterogeneous systems will have properties which cannot be obtained in simpler and more analytically tractable model systems. Perhaps now that many “pure” systems in OoL theories have been initially characterized, researchers can focus more on the properties of more “messy” chemical systems, including those containing both biomolecules and non-biomolecules.

Some studies suggest the modern complement of biomolecules was at least partially derived via natural selection (e.g., [27,82,83,84]). If so, the question of when and how pre-biomolecular processes, structures, or systems were replaced by the current set of biomolecules would be an important topic of investigation. For example, ribosomes are also able to catalyze polyester formation using tRNAs charged with αHAs instead of α-amino acids, suggesting that biological translation machinery is compatible with such non-biomolecule monomer substrates [85]. Although the process to generate αHA-charged tRNA in this study was through an artificial chemical process, the fact that αHA-charged tRNA could be generated suggests that αHA could have participated in ribosome-catalyzed elongation at some point in history possibly contemporaneously with ribozyme-catalyzed peptide elongation. Co-existence of αHA-charged and α-amino acid-charged tRNA may have resulted in a fairly promiscuous translation system, but eventually, α-amino acids and peptides dominated and thus some mechanism to remove this promiscuity would have been necessary.

While the promiscuity of the ribosome and other examples from biochemistry may be interpreted as vestigial [86], it is possible many, if not most, transitional mechanisms in the evolution of biochemistry have left no easily discernible trace in modern biology. As an analogy, the language the present manuscript is written in is itself a mashup of at least two major previously existing spoken and written languages (Old English and French), which themselves had predecessors that were also cross-seeded before being melded into English. These two languages, by the time of their merging, had themselves already been adapted to using alphabets derived from prior “dead languages” (meaning no one living at the time spoke or wrote them) that those literate in Old English or French also could not have understood. The analogy between human culturally transmitted knowledge and templated or non-templated chemical information transfer is obviously imperfect, but it points to the gaps in our knowledge and perhaps our imagination as to how such transitions may have occurred. This is especially true when the up to one billion years [87] available for the origins of life is considered.

Biological evolution is not teleological [88], and early chemical evolution could not have been either. Rather, each new chemical circumstance that arose from previous non-directed reactions created conditions that allowed for the creation of new reaction pathways, perhaps driven by changes in the environment. While we don’t know which exact geological process would have a role in the Origins of Life, Earth’s geological environment has changed over time (e.g., [89]). Assuming a diverse chemical milieu earlier on, with each subsequent geological change (such as air temperature, water salinity, mineral chemistry, atmosphere composition, etc.), some molecules/polymers that persisted, evolved, and/or functioned in an earlier geological state could either degrade or fail to function in the new geological state. At the same time, a select group of molecules could still persist in the new geological state. As the geological state changes again and again over time, this results in even more “pruning” of the total number of functional molecules in the system to result in a small set of functional molecules which persisted throughout all cycles of geological change. These geological transitions, assuming that geological changes were the only transitions on early Earth before the advent of living systems (once living (or life-like) systems emerged, then geological changes would not have been the only potential selective force on early Earth) thus led the way for the transition from the diverse prebiotic milieu into more selected and functional early biochemistries. As an example, one reasonable possibility is that early in Earth’s history, there was a much higher pCO_2_ atmosphere which would have resulted in considerably more acidic surface waters [90]. As esters are more stable than amides under acidic conditions, it is possible that as the pCO_2_ was lowered over time, the pH of surface waters increased, leading to a relatively greater stability of amides vs. esters as we see in present day biology. Such environmental or geological changes could also have led to innovation of increasingly specific and functional catalysts from the initially diverse and chaotic prebiotic milieu [91,92]. It is unclear whether the earliest catalysts were composed exclusively of biomolecules, or what catalytic functions primitive polymers would need to accomplish. Though it is complicated to study the abiological emergence of functional polymers due to uncertainties about the relevant environmental conditions and reactant availability, various screening and dynamic combinatorial techniques [24,93,94,95,96,97] may be useful to probe the properties of complex chemical systems. Nevertheless, understanding the potential contribution of the likely diversity of prebiotic organic chemistry on early Earth to contribute to life’s emergence is a worthy enterprise.

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
