# Peer review of "Polyesters as a Model System for Building Primitive Biologies from Non-Biological Prebiotic Chemistry"

_life, 2020, doi:10.3390/life10010006_

Round 1

Reviewer 1 Report

The paper constitutes an interesting case of how non-biological polymers could have supported and facilitated the emergence of life. The authors meticulously describe the diverse set of molecular monomer types that may have existed in a prebiotic setting based on hard data, and the sundry of chemical reactions they may display, in the realm of “messy chemistry”. They then explore the idea that some special features in this chemical mixture might have assisted, through chemical evolution, with the emergence of more biological systems by focusing on one type of chemical family, alpha hydroxy acids. The authors argue that droplets made of alpha hydroxy acids polymers are a suitable model for such chemical evolution, given their prebiotic prevalence, their labile nature and dynamics of formation, their encapsulation and catalytic capacities, and their ability to promote the rise of biologically–relevant chemistries, such as lipids, peptides and nucleic acids polymers. In an ecological view, Interactions between droplets, each having a specific chemistry, might have driven the process of natural selection towards more complex life. The paper is acceptable, with revisions as described below.

1) The paper has at its base the highly rational idea that life began in an environment that contained an extremely large excess of organic compound type, much beyond what is found in present-day living cells. The further point out that this vast “non-biological” repertoire could have significantly contributed to processes that led to life. For this purpose, they define in the introduction the term “biomolecules” for the compounds used in contemporary biochemistry and “non-biomolecules” for those that are not found in contemporary biochemistry, but could have been used in previous biological or proto-biological states. Strangely, they scarcely use their defined terms in what follows, and employ several other terms for the same definitions. Examples are lines 41, 84, 86, 90 and 109. This should be brought to order.

2) The introduction should convey a broader scope on life’s beginning with diverse non-biomolecules, beyond the mentioned alternative nucleic acids and polyesters (lines 90-93). Examples are: a) collectively autocatalytic sets of oligomers that may in principle be constructed of non-biomolecules [1], and b) self-assembling non-biomolecular lipid-like peptides that may have taken part in life’s emergence [2]. In a more general vein, I propose to mention the concept of “chemical opportunism” in life’s origin, which stresses that there is no compelling a priori reason to assume that life began with present-day biomolecules [3], with the published example of opportunistic proposed processes involving collectively autocatalytic set made solely of amphiphiles that are prebiotically available, and typically non-biomolecular [3, 4].

3) Since the authors stress the importance of taking into account the huge molecular diversity at life’s origin, the paper will benefit from explaining in more detail why they end up studying a very narrow molecular repertoire, restricted both by including very few hydroxyacids types and using only such whose residues are identical to those of biomolecular amino acids. It will be good to discuss what is expected to happen if these constraints were eased up.

4) Due to the utter heterogeneity of the prebiotic chemical environment each compound or small set of compounds is expected to be present at a very low concentration. Analyses of hydroxyacids in carbonaceous meteorites show that each is present at a few to several dozen nanomoles per gram sample [5]. If such 1 gram sample were to dissolve in 1 liter of water the total concentration of a few hydroxy acids would be at most 0.5 uM. This 1 million fold more dilute than the 0.5M concentration used to generate polyesters showing phase-separated droplets. Drying may lead to higher concentration for the hydroxyamino acids but so would happen with the huge excess of other compound types, with likely strong inhibition of “pure polymer” formation. The authors should discuss this sensitive point. Specifically it would be helpful to compare more promiscuous models that allow a large diversity of chemical compounds to take part in processes leading to life, to such that require a very specific subset, e.g. hydroxy acids.

5) The authors regard the phase-separated polyester droplets as protocells. Indeed, thermodynamic partition may lead to equilibrium differences in concentrations of various solutes between the two phases. It remains a bit vague what might be the non-equilibrium processes that the droplets may promote, in particular catalysis which is so essential for life’s emergence. Do the droplets described here contain catalytic branched polymers such as those alluded to in line 95, and if so, what might get catalyzed? Are there predictions for the establishment of catalytic networks?

6) The sentence in lines 117-119 “According to this model, genetic information transfer might have arisen among such systems, and then spread throughout them, co-opting and replacing their more primitive modes of reproduction” requires clarifications in ways that are related to the previous point. First, does “genetic information transfer” mean exchange of polynucleotides? And if so, what might they encode or catalyze and how would their across-droplet transfer make the droplets more life-like? Second, what is meant by “more primitive modes of reproduction”? The best candidate, I assume, is Kauffman-style collectively autocatalytic sets. If so, what inside the droplets would be the mutually catalytic molecules? And how will they be superseded by replicating polynucleotides? How would the content of the initial droplets possibly support reproducing polynucleotides?

7) The scenario in Figure 5 is a bit minimalistic, perhaps disappointing. Could the most important task for the droplets be serving as passive trash cans for poisonous B while the real life-approaching activity of A would take place in the external open milieu? It would be more convincing if at least one additional specific molecular scenario were brought up, assigning a more central mechanistic role to being enclosed within droplets.

References
** marks references already in the submitted manuscript

1) Kauffman, S.A., Approaches to the origin of life on earth. Life, 2011. 1(1): p. 34-48.

2) Zhang, S., Lipid-like self-assembling peptides. Accounts of chemical research, 2012. 45(12): p. 2142-2150. Lancet, D., R.

3) Zidovetzki, and O. Markovitch, Systems protobiology: origin of life in lipid catalytic networks. Journal of The Royal Society Interface, 2018. 15(144): p. 20180159.

4) **Kahana, A. and D. Lancet, Enceladus: first observed primordial soup could arbitrate origin of life debate. Astrobiology, 2019. .

5) **Pizzarello, S., Y. Wang, and G.M. Chaban, A comparative study of the hydroxy acids from the Murchison, GRA 95229 and LAP 02342 meteorites. Geochimica et Cosmochimica Acta, 2010. 74(21): p. 6206-6217.

Author Response

REVIEWER 1

The paper constitutes an interesting case of how non-biological polymers could have supported and facilitated the emergence of life. The authors meticulously describe the diverse set of molecular monomer types that may have existed in a prebiotic setting based on hard data, and the sundry of chemical reactions they may display, in the realm of “messy chemistry”. They then explore the idea that some special features in this chemical mixture might have assisted, through chemical evolution, with the emergence of more biological systems by focusing on one type of chemical family, alpha hydroxy acids. The authors argue that droplets made of alpha hydroxy acids polymers are a suitable model for such chemical evolution, given their prebiotic prevalence, their labile nature and dynamics of formation, their encapsulation and catalytic capacities, and their ability to promote the rise of biologically–relevant chemistries, such as lipids, peptides and nucleic acids polymers. In an ecological view, Interactions between droplets, each having a specific chemistry, might have driven the process of natural selection towards more complex life. The paper is acceptable, with revisions as described below.

We thank the reviewer for their comments and address their further concerns below.

1) The paper has at its base the highly rational idea that life began in an environment that contained an extremely large excess of organic compound type, much beyond what is found in present-day living cells. The further point out that this vast “non-biological” repertoire could have significantly contributed to processes that led to life. For this purpose, they define in the introduction the term “biomolecules” for the compounds used in contemporary biochemistry and “non-biomolecules” for those that are not found in contemporary biochemistry, but could have been used in previous biological or proto-biological states. Strangely, they scarcely use their defined terms in what follows, and employ several other terms for the same definitions. Examples are lines 41, 84, 86, 90 and 109. This should be brought to order.

We apologize for this oversight. We have carefully reinspected the manuscript and have standardized our nomenclature to reflect the reviewer’s comment.

2) The introduction should convey a broader scope on life’s beginning with diverse non-biomolecules, beyond the mentioned alternative nucleic acids and polyesters (lines 90-93). Examples are: a) collectively autocatalytic sets of oligomers that may in principle be constructed of non-biomolecules [1], and b) self-assembling non-biomolecular lipid-like peptides that may have taken part in life’s emergence [2]. In a more general vein, I propose to mention the concept of “chemical opportunism” in life’s origin, which stresses that there is no compelling a priori reason to assume that life began with present-day biomolecules [3], with the published example of opportunistic proposed processes involving collectively autocatalytic set made solely of amphiphiles that are prebiotically available, and typically non-biomolecular [3, 4].

We have added a short discussion about these systems as well as the notion of “chemical opportunism” in the introduction.

3) Since the authors stress the importance of taking into account the huge molecular diversity at life’s origin, the paper will benefit from explaining in more detail why they end up studying a very narrow molecular repertoire, restricted both by including very few hydroxy acids types and using only such whose residues are identical to those of biomolecular amino acids. It will be good to discuss what is expected to happen if these constraints were eased up.

This is a good point. The referenced paper (Jia, T.; Chandru, K; et. al. Membraneless polyester microdroplets as primordial compartments at the origins of life. Proc. Natl. Acad. Sci. USA. 2019, 116, 15830-15835) mostly functioned as a proof-of-principle demonstration of the polyester droplet assembly system from primitive hydroxy acid monomers. As such, the analytical tractability of the components present was of special concern in that study, and only pure solutions or simple mixtures of hydroxy acids that could form only linear polymers were chosen. We have added a short discussion about increasing the chemical diversity of polyester chemistries towards the end of the section titled “Self-Assembled Polyester Microdroplet Compartments.”

4) Due to the utter heterogeneity of the prebiotic chemical environment each compound or small set of compounds is expected to be present at a very low concentration. Analyses of hydroxyacids in carbonaceous meteorites show that each is present at a few to several dozen nanomoles per gram sample [5]. If such 1 gram sample were to dissolve in 1 liter of water the total concentration of a few hydroxy acids would be at most 0.5 uM. This 1 million fold more dilute than the 0.5M concentration used to generate polyesters showing phase-separated droplets. Drying may lead to higher concentration for the hydroxyamino acids but so would happen with the huge excess of other compound types, with likely strong inhibition of “pure polymer” formation. The authors should discuss this sensitive point. Specifically it would be helpful to compare more promiscuous models that allow a large diversity of chemical compounds to take part in processes leading to life, to such that require a very specific subset, e.g. hydroxy acids.

We agree with the reviewer’s point of view. We agree that studies of this type of heterogeneity are sorely lacking in the origins of life field. We have added a short discussion about this point and some references for prebiotic chemistry laboratory simulations which start to incorporate more diverse chemical systems stepwise (Forsythe, J.G.; Yu, S.-S.; Mamajanov, I.; Grover, M.A.; Krishnamurthy, R.; Fernández, F.M.; Hud, N.V. Ester-Mediated Amide Bond Formation Driven by Wet-Dry Cycles: A Possible Path to Polypeptides on the Prebiotic Earth. Angew. Chem. Int. Ed Engl. 2015, 54, 9871–9875. and Bhowmik, S.; Krishnamurthy, R. The role of sugar-backbone heterogeneity and chimeras in the simultaneous emergence of RNA and DNA. Nat. Chem. 2019, 1–10.) at the end of the section titled “Polyesters as a model system for chemical evolution” and at the beginning of the prospective section.

5) The authors regard the phase-separated polyester droplets as protocells. Indeed, thermodynamic partition may lead to equilibrium differences in concentrations of various solutes between the two phases. It remains a bit vague what might be the non-equilibrium processes that the droplets may promote, in particular catalysis which is so essential for life’s emergence. Do the droplets described here contain catalytic branched polymers such as those alluded to in line 95, and if so, what might get catalyzed? Are there predictions for the establishment of catalytic networks?

While the droplets in this discussion do not contain catalytic branched polymers, we are currently performing expanded experimental analyses of polyester droplets that incorporate more diverse chemistries and chemical functionalities, including branched polyesters. We have included a short discussion towards the end of the section titled “Self-Assembled Polyester Microdroplet Compartments.”

6) The sentence in lines 117-119 “According to this model, genetic information transfer might have arisen among such systems, and then spread throughout them, co-opting and replacing their more primitive modes of reproduction” requires clarifications in ways that are related to the previous point. First, does “genetic information transfer” mean exchange of polynucleotides? And if so, what might they encode or catalyze and how would their across-droplet transfer make the droplets more life-like? Second, what is meant by “more primitive modes of reproduction”? The best candidate, I assume, is Kauffman-style collectively autocatalytic sets. If so, what inside the droplets would be the mutually catalytic molecules? And how will they be superseded by replicating polynucleotides? How would the content of the initial droplets possibly support reproducing polynucleotides?

We have now clarified that genetic information transfer could have occurred through informational systems based on nucleic acids or other types of chemistry such as composomes. As mentioned in the previous point, we have already added further discussion on the possibilities of catalysis by polyester droplets in section titled “Self-Assembled Polyester Microdroplet Compartments”. At the end of the introduction, we have also added a more detailed explanation of proposed primitive systems which might function using horizontal genetic information transfer, i.e., progenotes, and their place in origins of life models.

7) The scenario in Figure 5 is a bit minimalistic, perhaps disappointing. Could the most important task for the droplets be serving as passive trash cans for poisonous B while the real life-approaching activity of A would take place in the external open milieu? It would be more convincing if at least one additional specific molecular scenario were brought up, assigning a more central mechanistic role to being enclosed within droplets.

We appreciate the reviewer’s point about such systems serving as “mere trash cans,” we have modified the text to underscore the point that this is but one of innumerable scenarios which one could explore. We have added text to this effect to the caption for Figure 5.

References

** marks references already in the submitted manuscript

1) Kauffman, S.A., Approaches to the origin of life on earth. Life, 2011. 1(1): p. 34-48.

2) Zhang, S., Lipid-like self-assembling peptides. Accounts of chemical research, 2012. 45(12): p. 2142-2150. Lancet, D., R.

3) Zidovetzki, and O. Markovitch, Systems protobiology: origin of life in lipid catalytic networks.Journal of The Royal Society Interface, 2018. 15(144): p. 20180159.

4) **Kahana, A. and D. Lancet, Enceladus: first observed primordial soup could arbitrate origin of life debate. Astrobiology, 2019. .

5) **Pizzarello, S., Y. Wang, and G.M. Chaban, A comparative study of the hydroxy acids from the Murchison, GRA 95229 and LAP 02342 meteorites. Geochimica et Cosmochimica Acta, 2010. 74(21): p. 6206-6217.

Reviewer 2 Report

Alfa-hydroxy acids were ubiquitous in carbonaceous meteorites and observed in many prebiotic syntheses systems. During a water-dry cycle, alfa-hydroxy acids form polyesters. This paper describes the role of polyesters in the prebiotic chemical evolution, in terms of compartmentation, encapsulation of molecules, and dynamic exchange of molecules. This paper is well-written and properly pointed out the main questions based on recent studies in this field. Here are several comments I have:

1) A general molecular structure of alfa-hydroxy acids and the reactions leading to the generation of polyesters should be added as a basic knowledge for the broad readers;

2) Although the polyesters can partly be considered to share a similar structure with polypeptides, there is clear difference between them. Polypeptides can fold due to hydrophilic, hydrophobic or electrostatic interactions. Can we expect similar intramolecular folding in the polyester system? (line 155) Is there any previous report on this? Please address this point.

3)The non-polar interior seems interesting (P7). It was suggested that these microreactors may be able to concentrate primitive chemical reactants, which is critical for many prebiotic syntheses. Can you explain in more detail the concentration mechanism?

4) There is another interesting idea on using these microdroplets to remove oligomerization inhibitors during template-directed non-enzymatic RNA polymerization in solution as shown in Figure 5. What is the challenging part of this scenario? How can a microdroplet selectively encapsulate B rather than A? What are the key physicochemical properties (sidechain, polarity or size, etc.?) of microdroplet required for this purpose?

5) Biological evolution is not teleological, however, they are strongly shaped by the geological environment (available energy sources and nutrients, etc.) Can you make a perspective on how the dynamic change of geological environment may have caused the selection or transition from a diverse, random prebiotic chemistry to selected biochemistry?

Author Response

REVIEWER 2

Alfa-hydroxy acids were ubiquitous in carbonaceous meteorites and observed in many prebiotic syntheses systems. During a water-dry cycle, alfa-hydroxy acids form polyesters. This paper describes the role of polyesters in the prebiotic chemical evolution, in terms of compartmentation, encapsulation of molecules, and dynamic exchange of molecules. This paper is well-written and properly pointed out the main questions based on recent studies in this field. Here are several comments I have:

We thank the reviewer for their comments, which have greatly helped to improve the quality of our manuscript.

1) A general molecular structure of alfa-hydroxy acids and the reactions leading to the generation of polyesters should be added as a basic knowledge for the broad readers;

We have added in more discussion about this, and a relevant figure, Scheme 1, in the appropriate section.

2) Although the polyesters can partly be considered to share a similar structure with polypeptides, there is clear difference between them. Polypeptides can fold due to hydrophilic, hydrophobic or electrostatic interactions. Can we expect similar intramolecular folding in the polyester system? (line 155) Is there any previous report on this? Please address this point.

We have included a brief discussion and some references about this topic toward the end of the introduction.

3)The non-polar interior seems interesting (P7). It was suggested that these microreactors may be able to concentrate primitive chemical reactants, which is critical for many prebiotic syntheses. Can you explain in more detail the concentration mechanism?

We have included a discussion and reference about partition coefficients (Tsai, W.-T. Environmental risk assessment of hydrofluoropolyethers (HFPEs). J. Hazard. Mater. 2007, 139, 185–192.) as the main factor which controls this property of compartmentalization systems toward the beginning of the section titled “Self-Assembled Polyester Microdroplet Compartments”.

4) There is another interesting idea on using these microdroplets to remove oligomerization inhibitors during template-directed non-enzymatic RNA polymerization in solution as shown in Figure 5. What is the challenging part of this scenario? How can a microdroplet selectively encapsulate B rather than A? What are the key physicochemical properties (sidechain, polarity or size, etc.?) of microdroplet required for this purpose?

This example is simply a thought experiment and has not been demonstrated in the laboratory, although we are currently exploring this. We now discuss the physicochemical properties that could allow selective encapsulation of B rather than A in the Figure 5 legend.

5) Biological evolution is not teleological, however, they are strongly shaped by the geological environment (available energy sources and nutrients, etc.) Can you make a perspective on how the dynamic change of geological environment may have caused the selection or transition from a diverse, random prebiotic chemistry to selected biochemistry?

This is a good point, and we have added a discussion, a reference (Hazen, R.M.; Filley, T.R.; Goodfriend, G.A. Selective Adsorption of L- and D-Amino Acids on Calcite: Implications for Biochemical Homochirality. Proc. Natl. Acad. Sci. USA 2001, 98, 5487–5490.), and a potential example in the prospective section to address this.

Reviewer 3 Report

The polyesters, a close descendant of present day protein like structures could be a prebiotic molecular scaffold for evolving the present day biomolecules. It has been an attractive field and getting more attention in past decades. The authors have summed up relevant scientific contributions in this broad field in a nice way. It could be published in its present form.  

Author Response

REVIEWER 3

The polyesters, a close descendant of present day protein like structures could be a prebiotic molecular scaffold for evolving the present day biomolecules. It has been an attractive field and getting more attention in past decades. The authors have summed up relevant scientific contributions in this broad field in a nice way. It could be published in its present form. 

We thank the reviewer for their positive comments.

Round 2

Reviewer 1 Report

No further comments

Reviewer 2 Report

The authors have properly responded to all the comments.